# Circulating inflammatory biomarkers and academic performance in adolescents: DADOS study

**Mireia Adelantado-Renau**[1], **Maria Reyes Beltran-Valls**[1], **Jorge Mota**[2], **Diego Moliner-Urdiales**[1]*

**1** LIFE research group, Universitat Jaume I, Castellón de la Plana, Castellon, Spain, **2** Research Center in Physical Activity, Health and Leisure (CIAFEL), Faculty of Sports at Porto University, Porto, Portugal

* dmoliner@uji.es

**Data Availability Statement:** All relevant data are within the paper and its Supporting information files.

## Abstract

### Objective

The present study aimed (1) to examine the association between circulating inflammatory biomarkers and academic performance in adolescents, and (2) to identify the ability of circulating inflammatory biomarkers to predict low academic performance.

### Methods

A total of 244 adolescents (13.9±0.3 years, 112 girls) from the DADOS study were included in the analysis. Four inflammatory biomarkers were quantified: white blood cell (WBC) count, interleukin-6, tumor necrosis factor-α (TNF-α), and C-reactive protein (CRP). Academic performance was assessed through academic grades and the Spanish version of the Science Research Associates Test of Educational Abilities.

### Results

TNF-α was inversely associated with math, Spanish and grade point average (β ranging from -0.166 to -0.124; all p<0.05), while CRP was inversely associated with verbal ability (β = -0.128; p<0.05). Overall, receiver operating characteristic (ROC) curves analyses showed discriminatory ability of WBC and TNF-α in identifying low academic performance (all p<0.05). Moreover, logistic regression analyses indicated that students with levels of WBC and TNF-α above the ROC cut-offs values showed between 78% to 87% increased likelihood of lower academic performance (p<0.05).

### Conclusions

Our findings suggested that some circulating inflammatory biomarkers were associated with academic performance in adolescents. Further larger longitudinal and interventional studies are needed to clarify the short-term and long-term relationship between inflammation and academic performance in youths.

**Funding:** The DADOS Study is funded by the Spanish Ministry of Economy and Competitiveness, MINECO (DEP2013-45515-R) and by the Jaume I University of Castellon, UJI (P1·1A2015-05 and UJI-A2019-12). This work was partly supported by a Sunny Sport research grant from the Schweppes Suntory Spain Company. M. A.R was supported by a Predoctoral Research Grant from UJI (PREDOC/2015/13 and E-2018-21). J.M was supported by grants: FCT: SFRH/BSAB/142983/2018 and UID/DTP/00617/2019, as well as Programa de Bolsas Santander Universidades 2018.

**Competing interests:** Schweppes Suntory Spain helped us to conduct the study providing their drinks during the study protocol (e.g., physical fitness tests). This does not alter our adherence to PLOS ONE policies on sharing data and materials.

## Introduction

Inflammation is a natural immune system response to injury, infectious agent, or oxidative stress. This mechanism can confer immune protection, promoting tissue survival, repair, and recovery. However, a prolonged activation of the peripheral immune system could lead to a state of systemic low-grade inflammation [1]. Prior research has suggested systemic low-grade inflammation to be both a cause, and a consequence of pathological processes related to several cardiovascular and metabolic diseases (e.g., atherosclerosis, diabetes, cancer) [2, 3], as well as to the development of neuropsychiatric disorders [4].

Emerging evidence indicates that circulating inflammatory biomarkers might also play a key role on cognition during the early and late stages of human lifespan [5, 6]. In fact, most of the studies reporting an inverse association between circulating inflammatory biomarkers and cognition have been focused on preterm infants [7, 8], and aging populations [9, 10], as well as in populations with neuropsychiatric disorders [11]. However, few studies have investigated the link between inflammation and cognitive function in adolescents, showing controversial results since not only negative [12], but also null [13] associations have been found.

Cognition may be closely linked to academic performance, which has shown to predict future health status [14] and work opportunities [15]. However, only one study has analysed the association between inflammation and academic performance in adolescents, showing an inverse association between circulating inflammatory biomarkers and academic grades [16].

Given the influence that inflammation may have on cognition, and the importance of academic performance for adolescents' future, it is of particular interest to clarify if inflammatory biomarkers are associated with academic performance in this age group. Thus, the present study aimed (1) to examine the association between circulating inflammatory biomarkers and academic performance in adolescents, and (2) to identify the ability of circulating inflammatory biomarkers to predict low academic performance.

## Materials and methods

### Study design and sample selection

The present work is part of the DADOS (Deporte, ADOlescencia y Salud) study, a research project aimed to analyse the influence of health-related factors on health and academic performance in adolescents [17]. The results presented in this study belong to baseline data obtained between March and May of 2015. A convenience sampling technique was used to recruit participants. For that purpose, advertising leaflets about the research project were sent to secondary schools and sport clubs located in Castellon (Spain). These included basic information and the general inclusion criteria of DADOS study which were: to be enrolled in 2nd grade of secondary school (i.e., the 8th grade), not having failed a previous academic year, and without having any medical diagnosis of physical or mental illness (as reported by participants' parents or guardians). Volunteers who met the inclusion criteria contacted the research group and were included in the study. We estimated that a sample of 300 participants would be required to provide statistical power of 80% with a level of significance of 0.05, assuming a dropout rate of 20%. Finally, from the total DADOS study sample (n = 274), 244 adolescents (112 girls) were included in the analyses. This final sample came from 38 secondary schools, out of 85 located in the province of Castellon (from which 19 were private schools, out of 34 located in this province), and had valid data for at least one circulating inflammatory biomarker and academic performance. All participants included in the current analyses self-reported before blood sample collection that they were not suffering from acute illnesses (e.g., flue, infection, fever, allergies, toothache) at the time of testing either during the last week.

Students and their parents or guardians were informed of the nature and characteristics of the study, and all provided written informed consent. The study protocol was designed in accordance with the ethical guidelines of the 1961 Declaration of Helsinki (last revision of Fortaleza, Brazil, 2013) and approved by the Research Ethics Committee of the Jaume I University of Castellon.

## Circulating inflammatory biomarkers

Blood samples were drawn from the antecubital vein after an overnight fast of at least 10 h (at 8:00 a.m.), and collected in two tubes containing EDTA (Greiner bioone, Kremsmünster, Austria). One tube was kept refrigerated at 4°C for immediate analyses in whole blood, while the other tube was centrifuged to obtain serum (3500 rpm for 10 min at 4°C). The following inflammatory biomarkers were quantified: white blood cell (WBC, $10^3/\mu L$) count, interleukin-6 (IL-6, pg/mL), tumor necrosis factor-$\alpha$ (TNF-$\alpha$, pg/mL), and C-reactive protein (CRP, mg/dL). WBC count was measured in whole blood by automated blood cell counters (ABX Pentra XL 80, Horiba ABX SAS; Montpellier, France) with an intra-assay precision coefficient of variation (CV) of <2%. IL-6 and TNF-$\alpha$ were determined in serum using specific sensitive Enzyme-Linked Immunosorbent Assay (ELISA) kits (DRG Instruments GmbH, Marburg, Germany) with a sensitivity of 2 pg/mL for IL-6, and 0.7 pg/mL for TNF-$\alpha$. The intra- and inter-assay precision CVs were 4.2% and 4.4% for IL-6, and 6.6% and 4.5% for TNF-$\alpha$, respectively. The CRP concentration was quantified in serum by immunoturbidimetry (CRP 981699, Thermo Fisher Scientific Oy; Vantaa, Finlandia) with a sensitivity of 6 pg/mL and intra- and inter-assay CVs of 2.6% and 0.8%, respectively.

## Academic performance

Academic performance was assessed using academic grades and a standardized test of academic abilities. Academic grades were taken from the participants' official report cards obtained at the end of the academic year, which were provided by parents or guardians. Individual grades for math, Spanish and English, as well as the grade point average score were included in the analyses. The grade point average score was calculated as the single average for geography and history, natural sciences, maths, Spanish, Catalan and English languages and physical education grades. All the subjects were measured on a ten-point scale, where 0 was the worst and 10 was the best. Academic abilities were assessed through the Spanish version of the Science Research Associates Test of Educational Abilities [18], which was completed within the same week that blood samples were collected. This test measures three basic academic abilities: verbal ability (command of language), numeric ability (speed and precision in performing operations with numbers and quantitative concepts), and reasoning ability (the aptitude to find logical ordination criteria in sets of numbers, figures, or letters). Scores for the three abilities were obtained by adding positive answers. Overall academic ability was calculated by adding the three abilities' scores (verbal + numeric + reasoning). The present study used level three, which is designed for adolescents aged 14–18 years. The alpha scores for its reliability have been reported to be 0.74 for verbal ability, 0.87 for numerical ability, 0.77 for reasoning ability and 0.89 for overall academic ability [18]. Participants were classified in high academic performance (≥50th of the median) and low academic performance (<50th of the median) for each academic performance indicator.

## Covariates

The statistical analyses were controlled for sex, pubertal stage, socioeconomic status related variables (i.e., parental educational level and type of school), waist circumference, and

adherence to the Mediterranean diet. These are relevant cofounders given the association of socioeconomic status [19], waist circumference [20, 21] and adherence to the Mediterranean diet [17, 22] with inflammation and academic performance. In addition, since adolescence is a period of developmental changes at a different pace, sex and pubertal stage were also considered as covariates.

**Pubertal stage.**   Pubertal stage was self-reported according to the five stages described by Tanner [23] based on the assessment of two components: pubic hair growth for boys and girls, plus breast development in girls, and genital development in boys. A 5-point maturity rating was used where stage 1 corresponds to the prepubertal state and stage 5 to mature state, and the highest rating of the two components was used for the analyses.

**Parental educational level.**   Parental educational level was used as a proxy of socioeconomic status [24]. Both parents reported their educational level and responses were combined as: neither of the parents had a university degree, and at least one of the parents had a university degree.

**Type of school.**   Students' school type was classified into 'public' or 'private' school, and entered as a dummy variable.

**Anthropometry.**   Measures were assessed in duplicate by experienced researchers following standardized procedures [25], and average measures were used for the analyses. Briefly, body weight was measured to the nearest 0.1 kg using an electronic scale (SECA 861, Hamburg, Germany). Height was measured to the nearest 0.1 cm using a wall-mounted stadiometer (SECA 213, Hamburg, Germany). Body mass index (BMI) was calculated as weight/height squared ($kg/m^2$). Participants were classified into normal weight and overweight or obese, according to the international age- and sex-specific BMI cut-offs proposed by Cole et al. [26]. Waist circumference was measured, as a proxy of abdominal obesity, to the nearest 1 mm with a non-elastic tape applied horizontally midway between the lowest rib margin and the iliac crest, at the end of gentle expiration with the adolescent in a standing position.

**Adherence to the Mediterranean diet.**   Adherence to the Mediterranean diet was evaluated using the KIDMED questionnaire, which includes 16 yes/no questions related to participants consumption of fast food, sweets and soft drinks, daily fruit and vegetables, and weekly fish and legumes [27]. Regarding the affirmative answers, a value of +1 was assigned to the questions with positive connotation in relation to Mediterranean diet (e.g., regular fruit consumption), while a value of -1 was assigned to the questions that constitute negative aspects (e.g., fast food consumption). Questions answered with "no" scored 0. The score for the students' level of adherence to the Mediterranean diet was calculated as the sum of each answer, ranging from 0 to 12.

## Statistical analysis

Descriptive characteristics of the study sample are presented as means ± standard deviation or frequency (%). Differences between sexes were examined by independent two-tailed t-tests and Chi-squared tests for continuous and categorical variables, respectively. All variables were checked for normality using both graphical (normal probability plots) and statistical (Kolmogorov-Smirnov test) procedures. Due to its skewed distribution, circulating inflammatory biomarkers were log-transformed when required. As preliminary analyses showed no significant interactions of sex with circulating inflammatory biomarkers in relation to academic performance indicators (all p>0.1), all analyses were performed with the total sample.

Linear regression analyses were used to study the association between circulating inflammatory biomarkers and academic performance indicators adjusting for sex, pubertal stage,

parental educational level, type of school, waist circumference and adherence to the Mediterranean diet.

Receiver operating characteristic (ROC) curves were conducted to investigate the ability of circulating inflammatory biomarkers (i.e., WBC, IL-6, TNF-α, and CRP) in discriminating low academic performance. The area under the curve (AUC) ranges between 0 and 1, where 0 represents a worthless test, and 1 a perfect ability of circulating inflammatory biomarkers to identify students with low academic performance. When the AUC was statistically significant, cut-off points were selected according to the highest Youden index, which is calculated with the best trade-off between sensitivity and specificity.

Based on the ROC curves analyses, logistic regression analyses were conducted to examine the relationships between high circulating inflammatory biomarkers concentrations (i.e., ≥cut-off values) and low academic performance, adjusting for sex, pubertal stage, parental educational level, type of school, waist circumference and adherence to the Mediterranean diet. These analyses were performed only for those circulating inflammatory biomarkers that showed a discriminatory ability to predict low academic performance (AUC>0.5 and p<0.05). All the analyses were performed using the IBM SPSS Statistics for Windows version 22.0 (Armonk, NY: IBM Corp), and the level of significance was set to p<0.05.

## Results

### Sample characteristics

Table 1 summarizes adolescents' characteristics by sex. Our study included 244 adolescents aged 13.9 ± 0.3 years old, of which 112 (45.9%) were girls. Boys presented higher values in height (164.7 vs. 160.9; p<0.001), waist circumference (68 vs. 66; p<0.01), TNF-α (5.6 vs. 4.8; p<0.01), and numeric ability (14.9 vs. 12.4; p<0.001) than girls.

### Association between inflammatory biomarkers and academic performance

Linear regression analyses between circulating inflammatory biomarkers and academic performance indicators after adjustment for sex, pubertal stage, parental educational level, type of school, waist circumference, and adherence to the Mediterranean diet are shown in Tables 2 and 3. WBC and IL-6 were not associated with academic performance indicators. TNF-α was negatively associated with math (β = -0.166; p<0.01), Spanish (β = -0.127; p<0.05), and grade point average (β = -0.124; p<0.05), while CRP was negatively associated with verbal ability (β = -0.128; p<0.05).

### Diagnostic ability of inflammatory biomarkers to predict low academic performance

Table 4 presents the parameters of the ROC curve analyses regarding the diagnostic ability of circulating inflammatory biomarkers to predict low academic performance in adolescents. ROC analyses showed that IL-6 and CRP concentrations did not discriminate among academic performance categories. However, significant AUC were found for WBC with English, and for TNF-α with all the academic grades indicators (all AUC>0.5 and p>0.05). No circulating inflammatory biomarker showed discriminatory ability to identify low academic abilities (S1 Table).

**Table 1. Descriptive characteristics for the study sample.**

| | All | Boys | Girls | p |
|---|---|---|---|---|
| n (%) | 244 (100) | 132 (54) | 112 (46) | |
| Age (y) | 13.9 ± 0.3 | 13.9 ± 0.3 | 13.9 ± 0.3 | 0.709 |
| Pubertal stage (I-V) (%) | 8/35/47/10 | 9/33/43/15 | 0/6/38/52/4 | - |
| Height (cm) | 163.0 ± 8.1 | 164.7 ± 8.6 | 160.9 ± 7.0 | **<0.001** |
| Weight (kg) | 53.8 ± 9.1 | 54.4 ± 9.3 | 53.1 ± 8.8 | 0.297 |
| Body mass index (kg/m$^2$) | 20.2 ± 2.6 | 19.9 ± 2.4 | 20.5 ± 2.8 | 0.106 |
| Overweight and obesity, n (%) | 29 (12) | 15 (11) | 14 (13) | 0.785 |
| Waist circumference (cm) | 67.1 ± 5.6 | 68.0 ± 5.2 | 66.0 ± 5.8 | **0.006** |
| Adherence to the Mediterranean diet (0–12) | 7.1 ± 2.1 | 7.3 ± 2.2 | 6.8 ± 2.1 | 0.059 |
| Parental educational level | | | | |
| University studies, n (%) | 118 (48) | 58 (44) | 60 (54) | 0.134 |
| School type | | | | |
| Private (%) | 67 (28) | 41 (31) | 26 (23) | 0.171 |
| Circulating inflammatory biomarkers [a] | | | | |
| White blood cells (10$^3$/μL) | 5.7 ± 1.4 | 5.6 ± 1.3 | 5.8 ± 1.4 | 0.426 |
| Interleukin-6 (pg/mL) (n = 206) | 3.1 ± 3.0 | 3.1 ± 3.1 | 3.1 ± 2.9 | 0.975 |
| Tumor necrosis factor-α (pg/mL) | 5.2 ± 2.2 | 5.6 ± 2.3 | 4.8 ± 2.1 | **0.004** |
| C-reactive protein (mg/dL) | 0.53 ± 0.21 | 0.54 ± 0.25 | 0.52 ± 0.14 | 0.312 |
| Academic grades (0–10) | | | | |
| Math | 6.53 ± 1.7 | 6.5 ± 1.7 | 6.7 ± 1.7 | 0.423 |
| Spanish | 6.7 ± 1.7 | 6.5 ± 1.7 | 7.0 ± 1.7 | **0.014** |
| English | 6.9 ± 1.7 | 6.7 ± 1.7 | 7.2 ± 1.7 | **0.025** |
| Grade point average | 6.9 ± 1.4 | 6.8 ± 1.3 | 7.0 ± 1.4 | 0.136 |
| Academic abilities | | | | |
| Verbal (0–50) | 19.0 ± 5.4 | 19.2 ± 6.0 | 18.8 ± 4.5 | 0.579 |
| Numeric (0–30) | 13.8 ± 4.7 | 14.9 ± 4.6 | 12.4 ± 4.4 | **<0.001** |
| Reasoning (0–30) | 16.7 ± 5.8 | 16.2 ± 5.7 | 17.3 ± 5.8 | 0.131 |
| Overall (0–110) | 49.5 ± 12.4 | 50.3 ± 13.0 | 48.5 ± 11.7 | 0.262 |

Data are presented as mean ± standard deviation or frequency (%). Differences between sexes were examined by t test or chi-square test. Statistically significant values are in bold.

[a] Values were log-transformed before analysis, but non-transformed values are presented.

**Table 2. Linear regression analysis examining the association between circulating inflammatory biomarkers and academic grades.**

| | Math | | | Spanish | | | English | | | Grade point average | | |
|---|---|---|---|---|---|---|---|---|---|---|---|---|
| | β | 95% CI | p | β | 95% CI | p | β | 95% CI | p | β | 95% CI | p |
| White blood cells | -0.072 | -3.395; 0.894 | 0.252 | -0.058 | -3.152; 1.140 | 0.357 | -0.108 | -3.888; 0.246 | 0.084 | -0.066 | -2.544; 0.742 | 0.281 |
| Interleukin-6 | 0.020 | -0.816; 1.110 | 0.765 | -0.042 | -1.297; 0.670 | 0.532 | 0.053 | -0.535; 1.263 | 0.426 | 0.009 | -0.692; 0.790 | 0.897 |
| Tumor necrosis factor-α | -0.166 | -2.201; -0.337 | **0.008** | -0.127 | -1.906; -0.032 | **0.043** | -0.112 | -1.752; 0.076 | 0.072 | -0.124 | -1.463; -0.027 | **0.042** |
| C-reactive protein | -0.068 | -4.192; 1.199 | 0.275 | -0.082 | -4.487; 0.894 | 0.190 | -0.087 | -4.470; 0.743 | 0.160 | -0.071 | -3.291; 0.835 | 0.242 |

β: standardized coefficient. CI: confidence interval. Analyses were adjusted for sex, pubertal stage, parental educational level, type of school, waist circumference, and adherence to the Mediterranean diet.

**Table 3. Linear regression analysis examining the association between circulating inflammatory biomarkers and academic abilities.**

| | Verbal | | | Numeric | | | Reasoning | | | Overall | | |
|---|---|---|---|---|---|---|---|---|---|---|---|---|
| | β | 95% CI | p | β | 95% CI | p | β | 95% CI | p | β | 95% CI | p |
| White blood cells | -0.041 | -8.716; 4.356 | 0.512 | -0.017 | -6.332; 4.712 | 0.773 | -0.007 | -7.651; 6.820 | 0.910 | -0.027 | -18.803; 11.993 | 0.664 |
| Interleukin-6 | 0.083 | -1.301; 5.104 | 0.243 | 0.105 | -0.508; 4.512 | 0.117 | 0.057 | -2.070; 4.834 | 0.431 | 0.100 | -2.082; 12.652 | 0.159 |
| Tumor necrosis factor-α | 0.020 | -2.481; 3.462 | 0.745 | -0.067 | -3.978; 1.042 | 0.250 | -0.096 | -5.856; 0.717 | 0.125 | -0.061 | -10.541; 3.446 | 0.319 |
| C-reactive protein | -0.128 | -17.224; -0.580 | **0.036** | 0.015 | -6.142; 8.068 | 0.790 | -0.058 | -13.685; 4.930 | 0.355 | -0.075 | -32.048; 7.415 | 0.220 |

β: standardized coefficient. CI: confidence interval. Analyses were adjusted for sex, pubertal stage, parental educational level, type of school, waist circumference, and adherence to the Mediterranean diet.

## Associations between high levels of inflammatory biomarkers and low academic performance

Fig 1 shows logistic regression analyses for the associations between high circulating inflammatory biomarkers (i.e., above the cut-off values provided by the ROC curve analysis) and low academic performance (i.e., <50th of the median) after adjustment for sex, pubertal stage, parental educational level, type of school, waist circumference, and adherence to the

**Table 4. Parameters of the receiver operating characteristic curve analysis for the diagnostic performance of circulating inflammatory biomarkers in identifying low academic grades.**

| Low academic grades | | White blood cells | Interleukin-6 | Tumor necrosis factor-α | C-reactive protein |
|---|---|---|---|---|---|
| Math | AUC | 0.499 | 0.496 | **0.582** | 0.536 |
| | 95%CI | 0.426–0.572 | 0.417–0.575 | 0.510–0.653 | 0.463–0.609 |
| | p | 0.982 | 0.929 | 0.028 | 0.329 |
| | Cut-off | - | - | ≥5.75 | - |
| | Sensitivity (%) | - | - | 0.713 | - |
| | Specificity (%) | - | - | 0.425 | - |
| Spanish | AUC | 0.549 | 0.491 | **0.604** | 0.505 |
| | 95%CI | 0.477–0.622 | 0.412–0.570 | 0.533–0.675 | 0.431–0.579 |
| | p | 0.186 | 0.822 | 0.005 | 0.893 |
| | Cut-off | - | - | ≥5.85 | - |
| | Sensitivity (%) | - | - | 0.732 | - |
| | Specificity (%) | - | - | 0.443 | - |
| English | AUC | **0.581** | 0.500 | **0.585** | 0.552 |
| | 95%CI | 0.508–0.653 | 0.420–0.580 | 0.513–0.658 | 0.479–0.626 |
| | p | 0.032 | 0.992 | 0.023 | 0.166 |
| | Cut-off | ≥5.25 | - | ≥5.85 | - |
| | Sensitivity (%) | 0.493 | - | 0.721 | - |
| | Specificity (%) | 0.654 | - | 0.442 | - |
| Grade point average | AUC | 0.562 | 0.514 | **0.598** | 0.506 |
| | 95%CI | 0.490–0.634 | 0.435–0.593 | 0.524–0.667 | 0.433–0.579 |
| | p | 0.094 | 0.727 | 0.010 | 0.872 |
| | Cut-off | - | - | ≥5.85 | - |
| | Sensitivity (%) | - | - | 0.727 | - |
| | Specificity (%) | - | - | 0.430 | - |

AUC: area under the curve; CI: confidence interval; Values in bold font indicate statistically significant AUC.

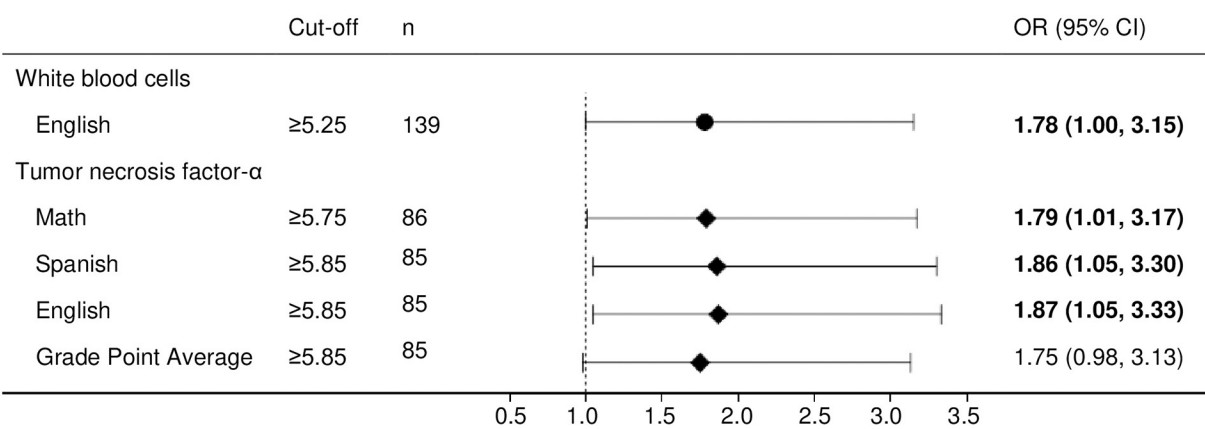

| | Cut-off | n | | OR (95% CI) |
|---|---|---|---|---|
| White blood cells | | | | |
| English | ≥5.25 | 139 | | **1.78 (1.00, 3.15)** |
| Tumor necrosis factor-α | | | | |
| Math | ≥5.75 | 86 | | **1.79 (1.01, 3.17)** |
| Spanish | ≥5.85 | 85 | | **1.86 (1.05, 3.30)** |
| English | ≥5.85 | 85 | | **1.87 (1.05, 3.33)** |
| Grade Point Average | ≥5.85 | 85 | | 1.75 (0.98, 3.13) |

**Fig 1. Logistic regression model predicting low academic grades according to high concentration of circulating inflammatory biomarkers (above the cut-off values provided by the ROC curve analysis).** Analysis adjusted for sex, pubertal stage, parental educational level, type of school, waist circumference, and adherence to the Mediterranean diet. Values in bold font indicate statistically significant results. OR: Odds ratio; CI: confidence interval. Reference (OR = 1.00): students with low concentration of circulating inflammatory biomarkers. n indicates number of adolescents from the total sample (n = 244) over the cut-off values.

Mediterranean diet. High levels of WBC were associated with low academic performance in English (OR = 1.78; 95%CI:1.00–3.15). In addition, students with high concentration of TNF-α had 79%, 86% and 87% increased odds of achieving low academic performance in math, Spanish, and English, respectively.

## Discussion

The main finding of the present study revealed an inverse association between TNF-α and math, Spanish, and grade point average after adjusting for potential confounders. In addition, CRP was inversely associated with verbal ability. Our study indicates that WBC presented discriminatory ability in identifying low academic performance in English, while TNF-α showed discriminatory ability in identifying low academic performance in all the academic grades analysed. Overall, students with high levels of WBC and TNF-α showed between 78% to 87% increased likelihood of low academic performance. These results further contribute to the scarce prior literature suggesting that inflammation may negatively influence academic performance in healthy adolescents [16].

To the best of our knowledge, no previous studies have examined the association between circulating inflammatory biomarkers and academic abilities, neither their ability for identifying low academic performance, which hampers comparisons among studies. To date, there is only one study that has investigated the association between inflammatory biomarkers and academic grades in adolescents [16]. In contrast to our results, Esteban-Cornejo et al. [16] found that WBC, IL-6 and CRP, but not TNF-α, were inversely associated with academic grades in math, Spanish, the mean of math and Spanish, and grade point average, independently of adiposity, in a sample of Spanish children and adolescents aged 6–18 years.

The reasons underlying the inverse association between TNF-α and academic grades as well as between CRP and verbal ability cannot be elucidated in the current study. However, these findings may be partially related to the key role that these inflammatory biomarkers play on brain functioning. Based on in vitro studies, we speculate that TNF-α could affect academic outcomes by reducing neurogenesis, increasing apoptosis of neurons, and consequently, inhibiting synaptic plasticity and memory consolidation [4]. Interestingly, although much work has

been carried out in mice, prior research in humans has suggested that TNF-$\alpha$-driven processes may contribute to cognitive impairments [6], which could also negatively influence academic results. In addition, a previous study indicated that CRP may negatively influence under-developed frontal brain regions involved in letter fluency-related skills [12], which in turn, might affect verbal ability. Moreover, based on results in adults, it is likely that CRP could alter academic performance through its negative effects on brain morphology and cognitive domains [28].

On the other hand, in the present study there was a lack of association between WBC and IL-6 with academic performance. Similarly, other previous interventional [29] and prospective [13] studies investigating the association between inflammation and cognitive function in youth have reported null findings. There are several hypotheses that could partially explain the lack of association found in our study. First, it is interesting to highlight that the lack of association found between IL-6 and academic performance indicators is generally consistent with previous research, suggesting that IL-6 does not affect proliferation and gliogenesis [4], which in turn, may have no influence on cognition or academic performance. Second, the fact that circulating inflammatory biomarkers at physiological levels can act with both, anti- and pro-inflammatory effects [10] could partially explain the divergent results found in the present study. Third, our sample showed optimal values of adherence to the Mediterranean diet and body composition, which have been related to lower levels of inflammatory biomarkers [21, 22]. Although in our sample levels of WBC and IL-6 are within the published range, they seem to be lower than the mean values reported in the existing literature [30, 31], which could partially explain that these concentrations did not affect students' academic performance. Lastly, the academic performance indicators included in the current study might not entirely capture the adverse effects that some circulating inflammatory biomarkers could have on cognition in adolescents.

The mixed results found in prior literature investigating the relationship of inflammation with cognition and academic performance in youths could be due to differences in participants' socioeconomic status [32], ethnicity [33] and lifestyles, as well as to methodological issues. In fact, divergent results could be related to the matrix (whole blood, serum or plasma, saliva) in which inflammatory biomarkers are measured in the studies [12], and even to the different technics of analysis implemented.

## Limitations and strengths

The results of the present study should be interpreted with caution. The cross-sectional design of our study does not allow us to draw any conclusion on the causal direction of the associations. In addition, the inclusion of a sample of apparently healthy adolescents limits the generalizability of our findings across the population. Likewise, it is plausible that the present study includes residual confounding from unmeasured variables such as direct diagnosis of acute illnesses (e.g., respiratory, gastrointestinal, or dental problems) from a physician. This is important since acute disease conditions, which are related to increased inflammatory proteins, could have been underestimated by participants. This issue, together with the fact that inflammatory levels were not evaluated several times, may have influenced the associations reported. Finally, multiple testing could involve an increase of the type I error rate (i.e., false positive). However, the strengths of the study comprise the use of different blood-derived inflammatory biomarkers, and the inclusion of a standardized test of academic abilities. In addition, our statistical analyses were controlled for pubertal stage, socioeconomic status related variables [19], waist circumference [20, 21] and adherence to the Mediterranean diet [17, 22], which are relevant given their association with inflammation and academic performance as suggested by previous research.

## Conclusions

In conclusion, our results suggested that some circulating inflammatory biomarkers were associated with academic performance in adolescents. Specifically, TNF-α was inversely associated with academic grades, while CRP was inversely associated with verbal ability. However, no association was found between WBC and IL-6 with adolescents' academic performance. Additionally, our results indicated that inflammatory biomarkers could be useful to identify students with higher risk of low academic performance. Since academic performance has shown to play a key role on future employability [15] and health status [14], further larger longitudinal and interventional studies are needed to clarify the short-term and long-term relationship between inflammation and academic performance in youths.

## Supporting information

**S1 Table. Parameters of the receiver operating characteristic curve analysis for the diagnostic performance of circulating inflammatory biomarkers in identifying low academic abilities.**
(DOCX)

## Author Contributions

**Conceptualization:** Mireia Adelantado-Renau, Maria Reyes Beltran-Valls, Diego Moliner-Urdiales.

**Data curation:** Mireia Adelantado-Renau, Diego Moliner-Urdiales.

**Formal analysis:** Mireia Adelantado-Renau, Maria Reyes Beltran-Valls, Diego Moliner-Urdiales.

**Funding acquisition:** Diego Moliner-Urdiales.

**Investigation:** Mireia Adelantado-Renau, Maria Reyes Beltran-Valls, Diego Moliner-Urdiales.

**Methodology:** Mireia Adelantado-Renau, Maria Reyes Beltran-Valls, Diego Moliner-Urdiales.

**Project administration:** Mireia Adelantado-Renau, Diego Moliner-Urdiales.

**Resources:** Diego Moliner-Urdiales.

**Supervision:** Mireia Adelantado-Renau, Diego Moliner-Urdiales.

**Validation:** Mireia Adelantado-Renau, Diego Moliner-Urdiales.

**Writing – original draft:** Mireia Adelantado-Renau.

**Writing – review & editing:** Mireia Adelantado-Renau, Maria Reyes Beltran-Valls, Jorge Mota, Diego Moliner-Urdiales.

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
