## [Decision Letter · Decision Letter 0]

20 Aug 2020

PONE-D-20-17331

CIRCULATING INFLAMMATORY BIOMARKERS AND ACADEMIC PERFORMANCE IN ADOLESCENTS: DADOS STUDY

PLOS ONE

Dear Dr. Moliner-Urdiales,

Thank you for submitting your manuscript to PLOS ONE. After careful consideration, we feel that it has merit but does not fully meet PLOS ONE’s publication criteria as it currently stands. Therefore, we invite you to submit a revised version of the manuscript that addresses the points raised during the review process.

What was the basis for considering sex, pubertal stage, parental educational level, type of school, waist circumference and adherence to the Mediterranean diet were included as covariates? Is there any evidence that variables like waist circumference and adherence to the Mediterranean diet affect the outcome of interest?In the study economic status related variables have not been measured and adjusted.  At least theoretically, low household economic may lead to both of inflammation and low academic performance and confound the relationship of interest. Why it was not possible to measure and adjust such variables in the study? Please state the statistical basis for reaching at the sample size of 244.Results: I don’t see any sub-section that summarizes the findings of the logistic regression analysis. Please include a short paragraph or so. Discussion: Please discuss limitation of the study including residual confounding from unmeasured variables and possible inflation of the type I error rate due to multiple testing. Its good that only students free from chronic diseases were included in the study. But it is not clear how the health status of the students was assessed. 

We look forward to receiving your revised manuscript.

Kind regards,

Samson Gebremedhin, PhD

Academic Editor

PLOS ONE

Additional Editor Comments:

Table 1: Some of the significant differences between the two genders are not identified (e.g. academic scores for English and Spanish).

Journal Requirements:

Could you therefore please include the title page into the beginning of your manuscript file itself, listing all authors and affiliations?

"The DADOS Study is funded by the Spanish Ministry of Economy and Competitiveness, MINECO (DEP2013-45515-R) and by the Jaume I University of Castellon, UJI (P1·1A2015-05; UJI-A2019-12).

This work is partly supported by a Sunny Sport research grant from the Schweppes Suntory Spain Company. MAR was supported by grant UJI E-2018-21. J.M was supported by grants FCT SFRH/BSAB/142983/2018 and UID/DTP/00617/2019, as well as Programa de Bolsas Santander Universidades 2018."

We note that you received funding from a commercial source: Schweppes Suntory Spain Company.

Reviewers' comments:

Reviewer's Responses to Questions

**Comments to the Author**

1. Is the manuscript technically sound, and do the data support the conclusions?

Reviewer #1: Partly

Reviewer #2: Partly

2. Has the statistical analysis been performed appropriately and rigorously? 

Reviewer #1: Yes

Reviewer #2: Yes

3. Have the authors made all data underlying the findings in their manuscript fully available?

Reviewer #1: Yes

Reviewer #2: Yes

4. Is the manuscript presented in an intelligible fashion and written in standard English?

Reviewer #1: Yes

Reviewer #2: Yes

5. Review Comments to the Author

Reviewer #1: The authors present a cross-sectional study of 244 boys and girls in order to test the hypothesis that inflammation mediators are associated with lower academic achievements.

The background information correctly argues for the feasibility of the hypothesis, and cite correct and up-to-date references.

Children were selected from both public and private schools, and all were on the same grade. The authors may wish to comment how many schools were approached and their possible representativeness in their community.

Little is shown about the actual recruiting process. The authors state the children were healthy and without chronic disease, they may wish to describe how this was evaluated. Furthermore, these mediators are also related to acute conditions; such as respiratory, gastrointestinal or dental problems, which may be more frequent in this age group. Some acute ailments may show seasonal variations during the year and it is possible to have several children affected at once. The authors may wish to comment on this.

To my knowledge inflammation mediators (IL-6 and TNF) were correctly assessed, CRP and leukocytes were also correctly measured (though, these are not mediators, but rather may show the effect of the mediators).

Academic performance was measured by both, grade point average on several subject maters and on a standarized test. It is important that the authors state if this test was applied at the same time of the measurement of mediators.

Covariates for the analysis were sex, pubertal development, parental educational achievement, type of school, and compliance with Mediterranean diet. Body mass index and waist circumference were also considered.

Logistic regression was performed with the value that best discriminated in ROC curve analysis; between achievement and the mediators. Since the values obtained with ROC curve analysis are close to the median of achievement (between 5 and 6); it would be important to know how these ratings relate to fail-pass grades in the Spanish educational system.

In the end TNF was associated to most of academic achievement ratings, and CRP was associated with verbal abilities ratings.

In the discussion, the authors explain their findings with previously published possible effects of both TNF and CRP in the central nervous system. Yet, the reasons for having higher mediators values are not fully explained. I believe that Mediterranean diet may not explain the full extent of these abnormalities. The authors may wish to comment on the degree of abnormality (i.e. compare these differences to normal values in their laboratory).

The greatest confounder in these circumstances tends to be obesity. ¿how many of their children were obese or overweight?, even if they used waist circumference in their logistic analysis.

The association makes perfect sense. Beyond the possible effects of TNF on the brain, If a child doesn’t feel well his/her performance will be substandard.

I believe the authors should comment on this aspects before the paper can be published.

Reviewer #2: - Please explain what method you have used to select your sample? e.g. randomly, convenience????

- Please explain how you select the schools.

Table 2. Please include 95% CI. You may exclude the none significant variables out of the table and explain those in the result?

Table 3. could you present table 3 in figure? may be the significant ones?

Minor comment: discussion - lines 211 - You may want to say the main finding of the present study reveals an association between TNF - alpha and math........ please fix similar wordings throughout.

Major comment.

We know that cognition as well as academic performance are results of various factors including nutrition, socio-economic factors, health status, genetic, environment etc or we can say long term effect of various factors. We also know that inflammation markers such as CRP and TNF alpha are early (maximum of 48 hrs) markers of inflammation or infection.

Now, my question is on the research question - are you trying to answer that inflammation is associated with cognitive and/or academic performance or you are tying to show that inflammation markers are associated with for instance, neurogenesis??? you have provided one in vitro-study that shows this relationship, is there any evidence from in vivo study that you could provide?

simple example. If the subjects were free of inflammation three days or more before data collection would the result be different?

Conclusion: should be based on your data.

6. PLOS authors have the option to publish the peer review history of their article (what does this mean?). If published, this will include your full peer review and any attached files.

Reviewer #1: No

Reviewer #2: No

---

## [Author Response · Author response to Decision Letter 0]

5 Oct 2020

Dear Samson Gebremedhin,

My co-authors and I are pleased to respond the editor’s and the reviewers’ comments and resubmit a revised version of the paper entitled: “Circulating inflammatory biomarkers and academic performance in adolescents: DADOS study” submitted as a Research article to PLOS ONE (No. PONE-D-20-17331). Changes to the original manuscript appear in blue text for ease of identification, and we believe that our manuscript is stronger as result of these modifications. An itemized point-by-point response to the journal requirements and the reviewers’ comments is attached to the resubmitted manuscript.

Yours sincerely,

The authors

---

## [Decision Letter · Decision Letter 1]

26 Oct 2020

CIRCULATING INFLAMMATORY BIOMARKERS AND ACADEMIC PERFORMANCE IN ADOLESCENTS: DADOS STUDY

PONE-D-20-17331R1

Dear Dr. Moliner-Urdiales,

We’re pleased to inform you that your manuscript has been judged scientifically suitable for publication and will be formally accepted for publication once it meets all outstanding technical requirements.

Kind regards,

Samson Gebremedhin, PhD

Academic Editor

PLOS ONE

Additional Editor Comments (optional):

Reviewers' comments:

Reviewer's Responses to Questions

**Comments to the Author**

1. If the authors have adequately addressed your comments raised in a previous round of review and you feel that this manuscript is now acceptable for publication, you may indicate that here to bypass the “Comments to the Author” section, enter your conflict of interest statement in the “Confidential to Editor” section, and submit your "Accept" recommendation.

Reviewer #1: All comments have been addressed

Reviewer #2: All comments have been addressed

2. Is the manuscript technically sound, and do the data support the conclusions?

Reviewer #1: Yes

Reviewer #2: Yes

3. Has the statistical analysis been performed appropriately and rigorously? 

Reviewer #1: Yes

Reviewer #2: Yes

4. Have the authors made all data underlying the findings in their manuscript fully available?

Reviewer #1: Yes

Reviewer #2: Yes

5. Is the manuscript presented in an intelligible fashion and written in standard English?

Reviewer #1: Yes

Reviewer #2: Yes

6. Review Comments to the Author

Reviewer #1: (No Response)

Reviewer #2: (No Response)

7. PLOS authors have the option to publish the peer review history of their article (what does this mean?). If published, this will include your full peer review and any attached files.

Reviewer #1: No

Reviewer #2: No

---

## [Editor Report · Acceptance letter]

29 Oct 2020

PONE-D-20-17331R1 

Circulating inflammatory biomarkers and academic performance in adolescents: DADOS study 

Dear Dr. Moliner-Urdiales:

I'm pleased to inform you that your manuscript has been deemed suitable for publication in PLOS ONE. Congratulations! Your manuscript is now with our production department. 

Kind regards, 

on behalf of

Dr. Samson Gebremedhin 

Academic Editor

PLOS ONE